# Short–Term Memory Convolutions

**Grzegorz Stefański, Krzysztof Arendt, Paweł Daniluk, Bartłomiej Jasik, Artur Szumaczuk**
Samsung R&D Institute Poland
{g.stefanski, k.arendt, p.daniluk, b.jasik, a.szumaczuk}@samsung.com

## Abstract

The real-time processing of time series signals is a critical issue for many real-life applications. The idea of real-time processing is especially important in audio domain as the human perception of sound is sensitive to any kind of disturbance in perceived signals, especially the lag between auditory and visual modalities. The rise of deep learning (DL) models complicated the landscape of signal processing. Although they often have superior quality compared to standard DSP methods, this advantage is diminished by higher latency. In this work we propose novel method for minimization of inference time latency and memory consumption, called Short-Term Memory Convolution (STMC) and its transposed counterpart. The main advantage of STMC is the low latency comparable to long short-term memory (LSTM) networks. Furthermore, the training of STMC-based models is faster and more stable as the method is based solely on convolutional neural networks (CNNs). In this study we demonstrate an application of this solution to a U-Net model for a speech separation task and GhostNet model in acoustic scene classification (ASC) task. In case of speech separation we achieved a 5-fold reduction in inference time and a 2-fold reduction in latency without affecting the output quality. The inference time for ASC task was up to 4 times faster while preserving the original accuracy.

## 1 Introduction

Convolutional neural networks (CNNs) arose to one of the dominant types of models in deep learning (DL). The most prominent example is the computer vision, e.g. image classification (Rawat & Wang, 2017), object detection (Zhao et al., 2019), or image segmentation (Minaee et al., 2021).

CNN models proved to be effective in certain signal processing tasks, especially where the long-term context is not required such as speech enhancement (Sun et al., 2021), sound source separation (Stoller et al., 2018), or sound event detection (Lim et al., 2017). Some authors showed that convolutional models can achieve similar performance that recurrent neural networks (RNN) at a fraction of model parameters (Takahashi & Mitsufuji, 2017). It is also argued that CNNs are easier to parallelize than recurrent neural networks (RNNs) (Gui et al., 2019; Liu et al., 2022; Rybalkin et al., 2021; Kong et al., 2021).

However, unlike RNNs, which can process incoming data one sample at the time, CNNs require a chunk of data to work correctly. The minimum chunk size is equal to the size of the receptive field which depends on the kernel sizes, strides, dilation, and a number of convolutional layers. Additionally, overlaps may be required to reduce the undesired edge effects of padding. Hence, standard CNN models are characterized by a higher latency than RNNs.

Algorithmic latency which is related to model requirements and limitations such as the minimal chunk size (e.g. size of a single FFT frame), a model look-ahead, etc. is inherent to the algorithm. It can be viewed as a delay between output and input signals under assumption that all computations are instantaneous. Computation time is the second component of latency. In case of CNNs it is not linearly dependent on the chunk size, due to the fact that the whole receptive field has to be processed regardless of the desired output size.

In the case of audio-visual signals, humans are able to spot the lag between audio and visual stimulus above 10 ms (Mcpherson et al., 2016). However, the maximum latency accepted in conversations can be up to 40 ms (Staelens et al., 2012; Jaekl et al., 2015; Ipser et al., 2017). For best human-device

interactions in many audio applications the buffer size is set to match the maximum acceptable latency.

## 1.1 RELATED WORKS

Many researchers presented solutions addressing the problem of latency minimization in signal processing models. Wilson et al. (2018) studied a model consisting of bidirectional LSTM (BLSTM), fully connected (FC) and convolutional layers. Firstly, they proposed to use unidirectional LSTM instead of BLSTM and found that it reduces latency by a factor of 2 while having little effect on the performance. Secondly, they proposed to alter the receptive field of each convolutional layer to be causal rather than centered on the currently processed data point. In addition, it was proposed to shift the input features with respect to the output, effectively providing a certain amount of future context, which the authors referred to as *look-ahead*. The authors showed that predicting future spectrogram masks comes at a significant cost in accuracy, with a reduction of 6.6 dB signal to distortion ratio (SDR) with 100 ms shift between input/output, compared to a zero-look-ahead causal model. It was argued, that this effect occurs because the model is not able to respond immediately to changing noise and speech characteristics.

Romaniuk et al. (2020) further modified the above-mentioned model by removing LSTM and FC layers and replacing the convolutional layers with depth-wise convolutions followed by point-wise convolutions, among other changes in the model. These changes allowed to achieve a 10-fold reduction in the fused multiply-accumulate operations per second (FMS/s). However, the computational complexity reduction was accompanied by a 10% reduction in signal to noise ratio (SNR) performance. The authors also introduced partial caching of input STFT frames, which they referred to as *incremental inference*. When each new frame arrives, it is padded with the recently cached input frames to match the receptive field of the model. Subsequently, the model processes the composite input and yields a corresponding output frame.

Kondratyuk et al. (2021) introduced a new family of CNN for online classification of videos (MoViNets). The authors developed a more comprehensive approach to data caching, namely layer-wise caching instead of input-only caching. MoViNets process videos in small consecutive subclips, requiring constant memory. It is achieved through so-called *stream buffers*, which cache feature maps at subclip boundaries. Using the stream buffers allows to reduce the peak memory consumption up to an order of magnitude in large 3D CNNs. The less significant impact of 2-fold memory reduction was noted in smaller networks. Since the method is aimed at online inference, stream buffers are best used in conjunction with causal networks. The authors enforced causality by moving right (future) padding to the left side (past). It was reported, that stream buffers lead to approximately 1% reduction in model accuracy and slight increase in computational complexity, which however might be implementation dependent.

## 1.2 NOVELTY

In this work we propose a novel approach to data caching in convolutional layers called Short-Term Memory Convolution (STMC) which allows processing of the arbitrary chunks without any computational overhead (i.e. after model initialization computational cost of a chunk processing linearly depends on its size), thus reducing computation time. The method is model- and task-agnostic as its only prerequisite is the use of stacked convolutional layers.

We also systematically address the problem of algorithmic latency (look-ahead namely) by discussing causality of the transposed convolutional layers and proposing necessary adjustments to guarantee causality of the auto-encoder-like CNN models.

The STMC layers are based on the following principles:

- Input data is processed in chunks of arbitrary size in an online mode.
- Each chunk is propagated through all convolutional layers, and the output of each layer is cached with shift registers (contrary to input-only caching as in Romaniuk et al. (2020)), providing so-called *past context*.
- The past context is never recalculated by any convolutional layer. When processing a time series all calculations are performed exactly once, regardless of the processed chunk size.

- The conversion of convolutional layers into STMC layers is performed after the model is trained, so the conversion does not affect the training time, nor does it change the model output.

STMC can be understood as a generalization of the stream buffering technique Kondratyuk et al. (2021), although it was developed independently with a goal to reduce latency and peak memory consumption in CNNs with standard, strided, and transposed convolutions as well as pooling layers (as opposed to standard convolutions only in MoViNets). STMC approaches the caching issue differently than MoViNets. It uses shift registers, which cache feature maps of all layers and shift them in the time axis according to the input size. The shift registers enable the processing of data chunks of arbitrary size, where the chunk size may vary in time. Furthermore, shift registers can buffer multiple states of the network, which allows for handling strides and pooling layers. Finally, we propose a transposed STMC, making it suitable for a broader family of networks, including U-nets.

## 2 METHODS

This section presents the Short-Term Memory Convolutions and their transposed counterpart.

### 2.1 SHORT-TERM MEMORY CONVOLUTIONS

The STMC and transposed STMC (tSTMC) layers introduce an additional cache memory to store past outcomes. Fig. 2 shows the conceptual diagram of our method including the interaction between introduced memory and actual convolution. The cache memory provides values calculated in the past to subsequent calls of the STMC layers in order to calculate the final output without any redundant computations. The goal is to process all data only once.

The design of the STMC shift registers is based on the observation that CNN layers of an online model process input data partially overlapping with data from the previous step. The only difference is that the new data is slightly shifted and only a small portion of the input is actually new. As shown in Fig 1A, assuming 2 convolutional kernels of size 2, at time step $t_1$ only the outputs concerning the new 6th frame were not already computed in time step $t_0$. In order to fully reconstruct the output without any redundant computations, the convolution results

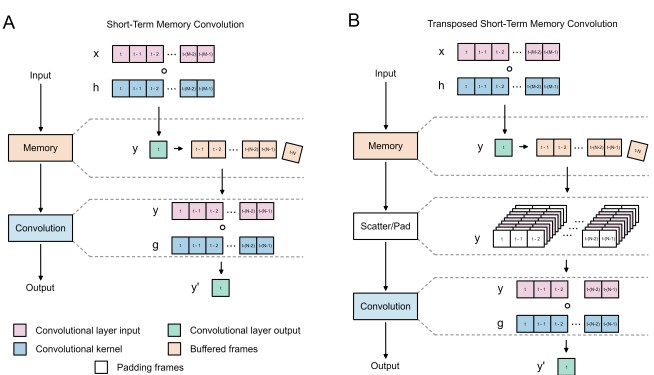

Figure 2: A) Short-Term Memory Convolution. B) Transposed Short-Term Memory Convolution.

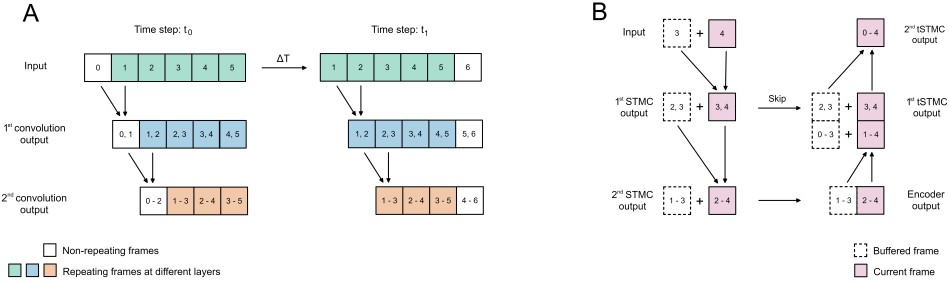

Figure 1: Effect of STMC on the network. A) Convolution redundancy effect in time domain. B) U-Net network with STMC/tSTMC.

(feature maps) must be saved at each level of the network (Fig. 1B). The number of saved frames in a particular STMC block is equal to the size of its receptive field. The input of the next convolutional layer is then padded with a previously saved state. This method retains all the past context from the network in an efficient way. The efficiency stems from the fact that all cached data is already pre-processed by the respective layer.

For the purpose of demonstration let us assume that a 1D time series data $X_t \in \mathbb{R}^N$ composed of $N$ samples is an input of the convolutional layer at time $t$. The convolutional layer has a kernel $h_1 \in \mathbb{R}^{M_1}$ which can be also represented by a Toeplitz matrix $H_1 \in \mathbb{R}^{N \times (N-M_1+1)}$ assuming *valid* padding. We can calculate the output signal $Y_t = (y_{t-N+M_1}, y_{t-N+M_1+1}, ..., y_t)$ as follows:

$$Y_t = \sigma(H_1 \cdot X_t^{\mathrm{T}}) \tag{1}$$

where $\sigma$ is an activation function applied element-wise. $Y$ is then processed by a second layer ($h_2 \in \mathbb{R}^{M_2}$ and $H_2 \in \mathbb{R}^{(N-M_1+1)\times(N-M_1-M_2+2)}$ respectively):

$$Z_t = \sigma(H_2 \cdot Y_t^{\mathrm{T}}) \tag{2}$$

Execution of the above operations for each subsequent time step $t$ has a computational complexity of $\mathrm{O}(NM_1) + \mathrm{O}(NM_2)$. To reduce the computational demand, one can apply the incremental inference scheme as described in (Romaniuk et al., 2020). Input data would be cropped to match the size of the receptive field, which in our example is $M_1 + M_2 - 1$. In case of $K$ layers of size bounded by $M$, computational complexity of incremental inference would be $\mathrm{O}(K^2 M^2)$.

STMC provides a further improvement, because it caches not only the inputs, but all intermediate outputs of all convolutional layers. These outputs are then used in a subsequent call of the model by concatenating them with the newest data. This technique ensures that all convolutions process only the newest data with kernels placed at the most recent position. Only the newest frames $y_t = \sigma(h \cdot X_t)$ need to be calculated:

$$Y_t = (Y_{t-1} \mid \sigma(h_1 \cdot X_t^{\mathrm{T}})) \tag{3}$$

where $X_t \in \mathbb{R}^{M_1}$ are the newest input frames and the $\mid$ represent the operation done by a shift register. For an arbitrary sequence $P$ and element $\alpha$ we have:

$$P = (p_1, p_2, ..., p_n) \tag{4}$$

$$(P|\alpha) = (p_2, p_3, ..., p_n, \alpha) \tag{5}$$

Thus $Z_t$ is computed as:

$$Z_t = (Z_{t-1}|\sigma(h_2 \cdot (Y_{t-1}|\sigma(h_1 \cdot X_t^{\mathrm{T}})))) \tag{6}$$

Complexity of this solution is $\mathrm{O}(M_1) + \mathrm{O}(M_2)$ or $\mathrm{O}(KM)$ for multiple layers under assumptions given above.

It is worth noting that the effectiveness of STMC increases with the depth of the network. Buffer $Y_{t-1}$ can be implemented as a simple shift register of size $M_1 - 1$ (Fig. 2). The method may be more complicated in practice depending on the architecture of the CNN. For example, strided convolutions require buffering of multiple states (using multiple shift registers). However, in this particular case, STMC would greatly improve the model's processing speed by mitigating stride-induced increased receptive field as the strided convolutions inflate the receptive field by the factor of stride. In comparison, incremental inference would be significantly affected by strided convolutions, because a larger receptive field requires more computations in the upper layers.

STMC can be applied to transposed convolutions as well. One approach is to use STMC with standard convolutions with zero-padding to facilitate upsampling. However, a more optimized approach exists, which we will refer to as *transposed STMC* (tSTMC). The Fig. 3 shows the diagram of our implementation of transposed convolution which can take advantage of caching and presents the partial outputs for each operation. It is worth noting that all pre-convolutional operators are applied only in frequency axis. Firstly, the scattering is applied to inflate the input by introducing zero-valued regions inside the input. Each region size depends on the stride of previous transposed convolution. Secondly, the padding is added to ensure desired output shape and then the convolution is applied.

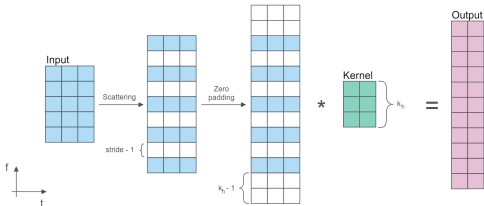

Figure 3: Transposed Short-Term Memory Convolution.

## 2.2 LIMITATIONS

STMC is a task-agnostic method and the presented caching and inference schemes can be applied to any task regardless of the domain. What is more STMC can be considered model-agnostic as well, because it can be applied to stacked convolutional layers anywhere in the model irrespectively of remaining modules.

As mentioned, the fist limitation of STMC is that it can be applied only to convolutional and pooling layers. In contrast to RNNs and LSTMs which take only the most recent data as inputs, convolutional models requires longer data segments as inputs. STMC simply caches the overlapping parts of the input segments with all intermediate results within the network between subsequent model calls.

The second limitation is that the exact implementation of STMC depends on the model's architecture. This work presents how to implement STMC to U-nets and GhostNet models. While it is possible to implement STMC to layers with strides and dilations, it would considerably increase the complexity of a network. E.g. strided convolutions require multiple shift registers per layer.

Finally, STMC should be used with causal convolutions, which typically have a lower latency than standard convolutions due to the lack of future context. In addition, causal convolutions are effectively "anti-causal" in the transposed convolutions, because the deconvolution output depends solely on the current and future input values (contrary to a causal case where output depends only on the past and the current input). This effect can be countered by shifting the model inputs with respect to model outputs, which works for stacked convolutional layers, but some additional modifications may be required in the case of more sophisticated architectures.

## 3 EXPERIMENTS

In this section we describe the experiments that evaluate the proposed method in terms of inference time and memory consumption. We present model architectures and experimental task alongside our dataset of choice and training setup. We provide the rationale behind the selected solutions.

### 3.1 SPEECH SEPARATION

#### 3.1.1 EXPERIMENTAL TASK

The speech separation is the main task we have selected to present the gains of STMC/tSTMC scheme. We adopt a general definition of this problem as the task of extracting broadly-understood speech signal from background interference. Because we are mainly focused on getting the clean speech signal the task can be understood as speech denoising or noise suppression. For clarity, we will refer to the stated task as a speech separation.

**Evaluation on desktop** The TFLite Model Benchmark Tool with C++ binary was adopted as an evaluation tool for inference time and peak memory consumption. Each model was tested in 1,000 independent runs on 8 cores of the Intel(R) Xeon(R) Gold 6246R CPU with 3.40 GHz clock, for which the average inference times and average peak memory consumption were calculated. Additionally, the average SI-SNRi metric was calculated to ensure the proposed optimization methods do not reduce the output's quality.

Latency $L$ was calculated using the following formula:

$$L = l_c + l_f + t_i + t_{\text{STFT}} + t_{\text{iSTFT}} \tag{7}$$

Components $l_c$ and $l_f$ comprise algorithmic latency: $l_c$ is the length of the current audio chunk, $l_f$ is the length of the look-ahead signal of the model – both are given in units of time. Components $t_i$, $t_{\text{STFT}}$ and $t_{\text{iSTFT}}$ relate to the computational cost: $t_i$ is the inference time, other two are time spent on calculating short-time Fourier transform and its inverse, respectively. Note that the length of future audio $l_f$ is zero in the case of causal models. STFT and iSTFT are computed for one frame only at each step. Our contribution demonstrates improvement in $t_i$ thanks to STMC as well as in $l_f$ due to introduced causality. For each model Latency was estimated using lowest possible length of input with which it is possible to achieve real-time processing, i.e. to have real-time factor (RTF) equal or smaller then 1.

**Evaluation on mobile device**    To confirm the portability of the proposed method to other devices, models were also evaluated on a mobile device using the same setup as on the desktop. We used Samsung Fold 3 SM-F925U equipped with Qualcomm Snapdragon 888. The models were analyzed using the TFLite Model Benchmark Tool, but this time 4 cores were used.

**STMC with different model sizes**    To provide some intuition on how STMC may perform with models of different sizes we applied STMC to the same U-Net architecture with a scaled number of kernels. We tested 16 models, which sizes range from 76 KB to 32 806 KB. To smooth out measured characteristics, each model size was trained 5 times and we report the average values for all our results. TFLite conversion for each model was done similarly to our main experiment.

**Influence of look-ahead**    As not every application requires such a small latency, STMC may still be introduced to reduce computational complexity. We provide the experimental results where we measured SI-SNRi of models using different lengths of look-ahead. We trained 5 models per size of look-ahead. Our tested look-ahead ranges from 0 to 7 frames with a step of 1 frame, where 0 is our causal network and 7 is our non-causal network from main experiment.

### 3.1.2 MODELS AND TRAINING PROCESS

The study is based on the U-net architecture as it consists of convolutions and transposed convolutions alongside skip connections. Because of that, it renders a greater challenge for STMC due to the "anti-causality" of the decoder, as described in Section A. What is more, it is widely use for solving audio domain tasks and it is proven to achieve very good results. In total, seven U-nets are compared: U-Net, U-Net$_{INC}$, U-Net$_{STMC}$, Causal U-Net, Causal U-Net$_{INC}$, Causal U-Net$_{STMC}$, and Causal U-Net$_{tSTMC}$. The models differ in causality and in STMC applied to the encoder and decoder. In addition, both U-Net and Causal U-Net process the whole data in a single inference, so they are effectively off-line models. Other models work in an online mode processing data in chunks.

**U-Net**    As our benchmark model, we incorporated a network based on the U-Net architecture (Ronneberger et al., 2015). The original architecture was simplified by removing max pooling operations, so the encoding is done solely by convolutions. We adapted the model for the audio separation task and achieved satisfying results with a model of depth 7. Each convolutional layer comprises a number of kernels of length 3 in the time domain, which leads to the receptive field of 29 for an assembled network. The kernel size in the frequency domain is set to 5. As the benchmark network consists of convolutional layers which are locally centered (Favero et al., 2021) on a current time frame, the receptive field of 14 for both past and the future context can be derived. The STFT input with real and imaginary parts as separate channels is utilized, resulting in a complex mask for the input signal as the output of the network. A STFT window of size 1024 and a hop of 256 are used.

**Causal U-Net**    To show the influence of causality we also trained Causal U-Net models. In all the considered causal models we used kernels of size 2 in the time domain to reduce the computation cost and the receptive field, as well as to compress the model size. The preliminary tests showed that the output quality of the casual model is not affected by the change. Such an approach allows to preserve only vital calculations and removes additional padding operations. The receptive field of the network is effectively reduced by half. To achieve causality, the output of the model was shifted by 7 STFT frames. To align the data inside of the network, all skip connections also needed to be shifted by the amount dependant on their depth inside the model.

**Incremental inference**    We applied the incremental inference method (Romaniuk et al., 2020) to U-Net and Causal U-Net by cropping the original input size to the size of receptive field, yielding 29 STFT frame input for U-Net and 15 STFT frames for the Causal U-Net. We will refer to this network as U-Net$_{INC}$ and Causal U-Net$_{INC}$.

**Short-Term Memory Convolutions**    Similarly to the incremental inference, STMC was applied to both U-Net and Causal U-Net networks. Firstly, we applied STMC inside the encoder and inside the decoder with zero-padding and output cropping (i.e. without tSTMC). We will denote these networks as U-Net$_{STMC}$ and Causal U-Net$_{STMC}$. To show the difference in inference time of STMC with cropping and with tSTMC, we also separately trained causal model consisting of both STMC and tSTMC. This network is denoted by Causal U-Net$_{tSTMC}$. The input size in the temporal axis for this method was set to 3 and 2 for U-Net and Causal U-Net, respectively.

**Dataset and training conditions**   The Deep Noise Suppression (DNS) Challenge - Interspeech 2020 dataset (Reddy et al., 2020), licensed under CC-BY 4.0, was adopted for both training and validation of networks as it can be easily processed for speech separation task. During training the models have seen around 100M randomly sliced 3-second audio samples. All models (U-Net, Causal U-Net and Causal U-Net$_{tSTMC}$) were trained on Nvidia A100 with batch size of 128 and Adam optimizer with learning rate of 1e-3. Each model was trained for 100 epochs which took about 90 hours per model. The further details about the training scheme and used losses were omitted because they do not affect neither the utility nor the effectiveness of the proposed method.

All models were converted into TFLite representation with default converter settings and without quantization.

## 3.2   Acoustic scene classification

### 3.2.1   Experimental task

To show that our method has also an impact in small CNN models we applied it to a GhostNet (Han et al., 2020) architecture trained for an acoustic scene classification (ASC) task. We tested the inference time and peak memory consumption on CPU (Intel(R) Xeon(R) Gold 6246R CPU with 3.40 GHz clock), GPU (Nvidia A100), and on CPU with XNNPACK optimization. For comparison, we also tested LSTM and MobileViT (Mehta & Rastegari, 2022) networks.

### 3.2.2   Models and training process

**GhostNet**   Our classification network consist of 3 ghost bottlenecks and ends with a convolutional layer and average pooling. The model is fed with a magnitude spectrogram of size 94 in the time axis and 256 in the frequency axis.

**LSTM**   We tested the RNN architecture with 2 LSTM blocks with 64 and 32 units and 2 fully connected layers with 16 and 1 units respectively. The model accepts an input of 1 time bin of magnitude spectrogram with 256 frequency bins.

**MobileViT**   We also implemented and trained the MobileViT network. In contrast to the original model described in (Mehta & Rastegari, 2022) we set the number of input channels to 1 and the number of output classes to 1.

**Dataset and training conditions**   The TAU Urban Acoustic Scene 2020 Mobile dataset (Heittola et al., 2020), was adopted for both training and validation of networks as our dataset for ASC task. We augmented the data by applying peak normalization, cropping and conversion to magnitude spectrograms in specfied order. We followed DCASE 2020 challenge task 1A instruction on train/test split. The further details about the training scheme and used losses were omitted because they do not affect neither the utility nor the effectiveness of the proposed method.

All models were converted into TFLite representation with default converter settings and without quantization.

## 4   Results

### 4.1   Speech separation

**Results on desktop**   The results for all models are presented side-by-side in Table 1. Comparing to standard convolutions, the incremental inference (INC) method achieves an 8.00-fold and a 29.74-fold latency reduction in the case of non-causal and causal setting, respectively. Our method (STMC) outperforms INC with a further 1.32-fold latency reduction for non-casual and a 2.52-fold reduction for causal variants. STMC achieved also approximately 5.68-fold reduction

Table 1: Results of evaluation on desktop using TFLite Model Benchmark Tool. 'C.' stands for causal. *Difference in metrics due to separate learning of models.

| Name | SI-SNRi (dB) | Inference time (ms) | Latency (ms) | Peak memory (MB) | Size (KB) |
|------|------|------|------|------|------|
| U-Net | **8.00** | 1 381.02 ± 17.62 | 4 403.22 | 170.36 | 14 025 |
| INC | **8.00** | 151.92 ± 12.25 | 550.54 | 48.97 | 14 025 |
| STMC | **8.00** | 86.33 ± 4.33 | 416.03 | 51.31 | 14 035 |
| C. U-Net | 7.58 | 962.61 ± 25.91 | 3 984.81 | 138.18 | **9 366** |
| C. INC | 7.58 | 55.64 ± 2.47 | 133.98 | 31.57 | 9 367 |
| C. STMC | 7.58 | 10.37 ± 0.69 | 52.07 | 27.50 | 9 376 |
| C. tSTMC | 7.44* | **9.80 ± 0.97** | **51.50** | **22.20** | 9 373 |

in inference time, compared to INC. The impact of STMC on the non-causal U-Net is lower as most of the computational cost of U-Net$_{STMC}$ is due to the need for waiting for the future context (convolutional kernels centered with respect to the output). The implementation of transposed STMC in the decoder of the Causal U-Net model gives only a slight improvement in latency (0.57 ms), but it reduces the peak memory consumption by approximately 20%. The model size is similar for standard, INC, and STMC variants. The causal models are approximately 33% smaller than the non-causal ones, which is caused by the smaller kernels as described in Section 3.1.2.

Table 2: Results of evaluation on mobile using TFLite Model Benchmark Tool. 'C.' stands for causal.

| Name | Inference time (ms) | Latency (ms) | Peak memory (MB) |
|---|---|---|---|
| U-Net | $262.44 \pm 26.81$ | 3 284.64 | 164.93 |
| INC | $38.51 \pm 4.34$ | 323.72 | 44.14 |
| STMC | $19.26 \pm 1.02$ | 288.96 | 46.59 |
| C. U-Net | $240.95 \pm 30.4$ | 3 263.15 | 133.60 |
| C. INC | $14.02 \pm 0.71$ | 50.74 | 26.53 |
| C. STMC | $3.15 \pm 0.10$ | 28.85 | 21.54 |
| C. tSTMC | $\mathbf{2.77 \pm 0.26}$ | $\mathbf{28.47}$ | $\mathbf{16.11}$ |

**Results on mobile device** The results of mobile evaluation are collected in Table 2. They further confirm the conclusions drawn from desktop results. Since the models used in this test are the same as in the case of desktop evaluation, we did not include the model size and SI-SNRi metrics in the table.

The inference time and latency achieved on the mobile device are significantly lower than on the desktop, despite the same evaluation method used on both platforms. The inference time reduction factor between Causal U-Net$_{INC}$ and Causal U-Net$_{tSTMC}$ is 5.06. The usage of tSTMC on this device has a relatively higher impact than on the desktop, as in this case it reduces the inference time by 12.06% (compared to 5.50%). Similarly to the desktop, the lowest peak memory consumption was achieved by Causal U-Net$_{tSTMC}$.

**STMC with different model sizes** The results for this set of experiments are displayed in fig. 4 by a set of 4 plots. As it was highlighted before, both INC and STMC models have the same SI-SNRi metrics and the dependency of this metric on model size is showed in fig. 4A. Absolute peak memory consumption improvement from STMC scales linearly with model size (fig. 4B), although the relative improvement compared to INC linearly decreases from 16% to 11%. STMC has significant influence on inference time in all tested models which can be seen in fig. 4C-D. For both inference patterns, similarly to absolute peak memory improvement, inference time seems to scale linearly with model size (fig. 4C). Speed-up of STMC over INC for this architecture is constant and amounts to around 81%, but drops for models of size 1.32 MB and smaller to around 74%.

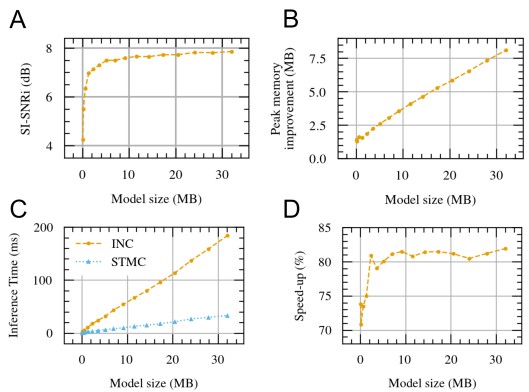

Figure 4: Influence of model size on STMC. A) SI-SNRi. B) Improvement over peak memory consumption. C) Inference time. D) STMC speed-up over INC.

**Influence of look-ahead** As can be seen in fig. 5 the look-ahead has a positive impact on metrics. It should also be noted that this experiment confirms that the 6% decrease of metrics is attributed exclusively to the lack of future content as in this test we actually interpolated between our causal and non-causal networks from main experiment. Determined characteristic is not linear and the first frames seems to be more significant than the furthest to the right future frames ones as first frame increases SI-SNRi by 0.177 dB and the last one by 0.019 dB. In this plot we included error bars as the difference between the same models is more significant than in every other experiment we presented in this work. Within error bars for specific look-ahead falls all SI-SNRi results on validation dataset we achieved for all 5 models of particular look-ahead we trained.

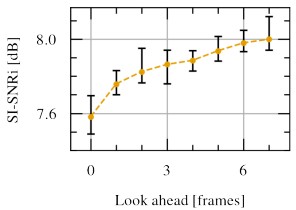

Figure 5: Influence of look-ahead on SI-SNRi.

## 4.2 ACOUSTIC SCENE CLASSIFICATION

Applying STMC to the Ghost-Net architecture has no impact on accuracy metrics. In addition we observed 4.84-fold reduction of CPU inference time, 34% improvement of inference time for GPU and 2.24-fold reduction for XNNPACK. In case of peak memory consumption we observed a 22% reduction for CPU and 30% reduction for XNNPACK with 4 KB increase in model size. We do not report peak memory consumption for GPU inference be-

Table 3: Results of ASC experiment.

| Network | Top-1 Acc (%) | Inference time (ms) | Peak memory (MB) | Number of parameters | Input size |
|---|---|---|---|---|---|
| GhostNet | | $4.706 \pm 6.167$ | 9.76 | | |
| GhostNet$_{GPU}$ | 65.83 | $0.912 \pm 0.504$ | — | 8 296 | $94 \times 256 \times 1$ |
| GhostNet$_{XNNPACK}$ | | $1.086 \pm 5.612$ | 11.49 | | |
| GhostNet$^{STMC}$ | | $0.973 \pm 0.866$ | 7.59 | | |
| GhostNet$^{STMC}_{GPU}$ | 65.83 | $0.602 \pm 0.321$ | — | 8 300 | $1 \times 256 \times 1$ |
| GhostNet$^{STMC}_{XNNPACK}$ | | $0.484 \pm 2.447$ | 8.10 | | |
| LSTM | | $0.095 \pm 0.010$ | 7.50 | | |
| LSTM$_{GPU}$ | 60.84 | $0.099 \pm 0.074$ | — | 95 137 | $1 \times 256$ |
| LSTM$_{XNNPACK}$ | | $0.119 \pm 0.348$ | 7.70 | | |
| MobileViT XXS | 69.58 | $52.948 \pm 11.456$ | 38.11 | 1 306 049 | $256 \times 256 \times 1$ |
| MobileViT XXS$_{GPU}$ | | $419.203 \pm 90.989$ | — | | |

cause the tool we used does not provide this information. By comparing STMC GhostNet to the LSTM we can observe that inference time of LSTM is up to 10x faster but comes at the cost of 12x higher number of parameters. It is important to note that the LSTM's supremacy of inference time over STMC decreases to 4x faster inference for XNNPACK. Additionally, our tests showed that CNNs were overall more unstable in case of inference time than LSTMs. Accuracy of LSTM model was at 60.84% which is lower than STMC GhostNet with accuracy of 65.83%. The MobileViT, at 1.3M parameters, scored the highest observed accuracy of 69.58% alongside the highest inference time of 52 ms. This model got the best metrics for the given ASC task but high inference time discredits its use for online processing. The best performing model on TAU dataset which was submitted in the DCASE 2020 challenge task 1A had an accuracy of of 74.4%, thus all models in our study can be deemed viable.

## 5 CONCLUSION

In this paper, we presented the method for inference time, latency, and memory consumption reduction. It relies on buffering of layer outputs in each iteration. The overall goal is to process all data only once and re-use past results of all convolutional layers. The method was tested on an speech separation task on two devices: desktop computer and mobile phone and ASC task with Ghost-Net on x86 CPU and A100 GPU. In total for audio separation, 7 variants of a U-Net architecture were compared, differing in causality and buffering types. On both x86 and ARM, approximately a 5-fold reduction of inference time was achieved compared to the incremental inference, which is considered the state-of-the-art method for online processing using convolutional models. The STMC method has also 44-61% lower latency and 30-39% lower peak memory consumption, depending on the test platform. All the causal models used in the test achieved approximately 6% reduction in SI-SNRi compared to their non-causal counterparts. The cause of the score reduction is attributed to the lack of future context and not the used caching method.

The STMC method proved to significantly reduce the number of operations inside CNNs, which may be crucial for many DL applications, e.g. deploying models on embedded devices. The method can be applied to most of the existing CNN architectures and can be combined with several different methods used for time series processing, e.g. dilated causal convolutions (van den Oord et al., 2016) or temporal convolutions (Lea et al., 2017). Using STMC and their transposed counterpart – tSTMC – does not require any retraining of the model. It is, however, strongly advised to use causal models in order to reach near real-time processing.

The described approach is also relevant for applications beyond the audio separation task presented in this work. Since the STMC method leads to the reduction of inference time, latency, and peak memory consumption, it can be used for systems where the size of data is too large for the available memory or computational power, e.g.: medical imaging (Blumberg et al., 2018), radio astronomy imaging (Corda et al., 2020), FPGA (Ma et al., 2018), video analysis (Lea et al., 2017), checksum computation (Filippas et al., 2022). We also believe that the described method may be useful for low latency systems on chip (SoC) for online processing applications (Park et al., 2022).

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

## A  EFFICIENT CAUSALITY

Non-causal convolutions can be used in online inference by either waiting enough time for their receptive field to be filled with "future" data, which increases latency, or by replacing the future data with zeros or some other values, e.g. symmetrically reflecting the data. The latter approach, however, degrades output quality at the most recent frames, which coincidentally are the most important for the online inference. Therefore, causality is typically required for robust online inference using convolutional networks. Causality is also desired for STMC, because STMC layers cache the past context as described in Section 2.1.

Technically, the convolution operation does not take any assumptions on causality. In many applications, especially in computer vision, kernels are centered with respect to the current output element of interest. For example, let $C$ be a convolutional layer with $3 \times 1$ dimensional kernel and valid padding. Let $A$ and $B$ be sample input and output tensors with dimensions of $5 \times 1$ and $3 \times 1$ respectively. We then have:

$$CA^T = \begin{bmatrix} k_1 & k_2 & k_3 & 0 & 0 \\ 0 & k_1 & k_2 & k_3 & 0 \\ 0 & 0 & k_1 & k_2 & k_3 \end{bmatrix} \begin{bmatrix} a_1 \\ a_2 \\ a_3 \\ a_4 \\ a_5 \end{bmatrix} = \begin{bmatrix} b_2 \\ b_3 \\ b_4 \end{bmatrix} = B^T \tag{8}$$

When the convolution is used this way, both the past and future context is needed.

One of the methods for enforcing causality is by using masked convolutions, in which the kernels are multiplied by non-symmetric masks (Van Den Oord et al., 2016) before being convolved with the input. However, masked convolutions are more resource consuming than standard convolutions due to additional multiplications. Therefore, in signal processing it is more desired to enforce causality by shifting the input with respect to the output (van den Oord et al., 2016). In other words, the elements of tensor $B$ can be re-indexed to reflect temporal dependency in a way that $b_i$ does not depend on any $a_j$ for $j > i$:

$$CA^T = \begin{bmatrix} k_1 & k_2 & k_3 & 0 & 0 \\ 0 & k_1 & k_2 & k_3 & 0 \\ 0 & 0 & k_1 & k_2 & k_3 \end{bmatrix} \begin{bmatrix} a_1 \\ a_2 \\ a_3 \\ a_4 \\ a_5 \end{bmatrix} = \begin{bmatrix} b_3 \\ b_4 \\ b_5 \end{bmatrix} = B^T \tag{9}$$

$$b_i = k_1 a_{i-2} + k_2 a_{i-1} + k_3 a_i \tag{10}$$

Our case study also requires the model to reconstruct a task-specified version of an input signal from its convolved representation. The simplest and effective approach to obtain such an inverse of the convolution is to perform transpose-like operation on its Toeplitz matrix, as depicted in equation (11). Note that convolution and transposed convolution do not share kernel parameters, but their matrices are in transpose-relation with respect to shape and structure.

An interesting phenomenon occurs when a "causal" convolution is transposed, because we then get:

$$C'B^T = \begin{bmatrix} k'_1 & 0 & 0 \\ k'_2 & k'_1 & 0 \\ k'_3 & k'_2 & k'_1 \\ 0 & k'_3 & k'_2 \\ 0 & 0 & k'_3 \end{bmatrix} \begin{bmatrix} b_3 \\ b_4 \\ b_5 \end{bmatrix} = \begin{bmatrix} a'_1 \\ a'_2 \\ a'_3 \\ a'_4 \\ a'_5 \end{bmatrix} = A'^T \tag{11}$$

$$a'_i = k'_3 b_i + k'_2 b_{i+1} + k'_1 b_{i+2} \tag{12}$$

which is effectively "anti-causal", i.e. the deconvolution output depends solely on current and future input values (contrary to a causal case where output depends only on past and current input). This effect can be countered by shifting model inputs with respect to model outputs:

$$C'B^T = \begin{bmatrix} 0 & 0 & 0 \\ 0 & 0 & 0 \\ k'_1 & 0 & 0 \\ k'_2 & k'_1 & 0 \\ k'_3 & k'_2 & k'_1 \end{bmatrix} \begin{bmatrix} b_3 \\ b_4 \\ b_5 \end{bmatrix} = \begin{bmatrix} a'_1 \\ a'_2 \\ a'_3 \\ a'_4 \\ a'_5 \end{bmatrix} = A'^T \tag{13}$$

$$a'_i = k'_3 b_{i-2} + k'_2 b_{i-1} + k'_1 b_i \tag{14}$$

This approach works for stacked convolutional layers, but in the case of more sophisticated architectures some modifications may be required. For example, U-Net models have skip connections between convolutions and transposed convolutions. The data passed through each skip connection must also be properly shifted with respect to the outputs of transposed convolutions. The shift value depends on the network architecture and may be different at each level.

