# OpenReview forum: "Short-Term Memory Convolutions"
_ICLR.cc/2023/Conference — ICLR 2023 poster_

### Official Review · Reviewer_EphW · 2022-10-25

**Confidence:** 2
**Correctness:** 3
**Technical Novelty And Significance:** 2
**Empirical Novelty And Significance:** 2
**Recommendation:** 6

**Clarity, Quality, Novelty And Reproducibility:**

Clarity: Some parts are not clear or confusing. For example, the application domain is not clear (denoising or speech separation), there is a mention of quantization in the appendix but not in the main text. In Table 3, are there any missing numbers or what do the centered numbers mean?

Quality: Fair. Experiments, especially the causal convolution based ones show significant reduction in inference time but it comes with a 6% degradation of SI-SNRi. Is there a way to avoid this trade-off?

Novelty: The model proposes to use a shift-register to cache the intermediate convolution outputs at intermediate layers to reduce the re-computation time.

Reproducibility: Some experimental details are provided, it should be reproducible at a high level but some details are missing. For example, whether there is model quantization, how the inference time processing is performed exactly: What is meant by ``Input data is processed in chunks of arbitrary size in an online mode.’’?

After the rebuttal phase, the paper has improved and got clearer. Hence I am increasing my initial score to 6.

Other comments/questions:
- What is the task? Is it denoising or speech separation? From the dataset (DNS), it sounds like the former but in several paragraphs it is told to be separation.
- In Appendix A, the equation for $b_i$ does not match with the matrix shown above it. Should the filter be reversed in the matrix $[k_3, k_2, k_1, 0, 0]$?
- Why are the inference time patterns different between Fig 4D and 4E?
- In Appendix B, there is a mention of quantization but the main paper does not mention anything related to this. Does the paper apply model quantization in the experiments of the main text?
- In the last sentence of the conclusion, it is mentioned that the SI-SNRi degradation in the causal STMC version is attributed to the lack of future context. Could the training vs. inference time mismatch be another reason?


**Strength And Weaknesses:**

Strengths:
- Experimental results show that the proposed model reduces the inference time and memory requirements.
- Some limitations of the proposed model have been clearly mentioned.

Weaknesses:
- The application domain is not clear. Even though the paper mentions audio separation several times, it uses deep noise suppression challenge data which is mainly used for noise suppression. In addition, independent of the application, the paper does not mention the loss function and what the outputs from the model are.

- The transposed version of the STMC model is not well-described. For example, in Fig. 2B, from where do we get the [0,1] into the buffer in the rightmost part of the figure?

- Figures are not described in detail in the main text.

**Summary Of The Paper:**

This paper proposes a method to store convolution outputs that have been already seen for the past data in a shift register and thus tries to avoid recomputation of the convolution output from those samples and works with the newly arrived data for online inference in convolutional neural nets. The final goal is to reduce the overall inference time. In the experiments, deep noise suppression (DNS) data is used in order to demonstrate the speed of the proposed model in a U-net architecture. Experiments compare causal and non-causal convolutions, their short-term memory (STMC) implementation and combination of STMC with transposed STMC (tSTMC).

**Summary Of The Review:**

The model introduces the use a shift-register to cache the intermediate convolution outputs at intermediate layers to reduce the inference time of convolutional networks. The method is applied only at the inference time and not during training. Experimental results show significant speed up in inference time with a reduced memory requirement. However, the best performing version of the model also degrades the signal quality. There is a trade-off between time vs. quality. The paper can get confusing at times (e.g., what is the task?). Analysis at the appendix could have been added to the main discussion.

---

> ### Author Response · Authors · 2022-11-18
> **Response 1/3**
>
> Hello! As all of reviews we received were extremely useful and helped us improve our paper, we would like to thank you for your time and insight. We are happy to say that the corrected version of the paper is ready but before you have a look please let us answer all of your questions and concerns.
>
> > The application domain is not clear. Even though the paper mentions audio separation several times, it uses deep noise suppression challenge data which is mainly used for noise suppression. In addition, independent of the application, the paper does not mention the loss function and what the outputs from the model are.
>
> The method presented in the paper is model and task agnostic as it is applicable to any model using a stack of convolutional layers (possibly including transposed convolutions). Because of that we did not focus on the training scheme or the used loss function. The main task used to illustrate our method in the paper is a speech separation (which is equivalent to speech enhancement/denoising, because the other sources are considered as noise). Additionally, we have also provided results for audio classification. The DNS dataset was used for training because it consists of clean speech samples and separated noise audio that can be conveniently used as target separated audio samples while their mix can serve as the model input. It is worth noting, that STMC does not affect the results. That’s is why we focused mostly on the latency and peak memory consumption and not on a particular use case.
>
> > The transposed version of the STMC model is not well-described. For example, in Fig. 2B, from where do we get the [0,1] into the buffer in the rightmost part of the figure?
>
> [0,1] was an error, it should be [0-3]. Thank you for pointing that out! This will be reflected in the corrected version of the paper. The frame [0-3] was a part of previous inference (the same case as in standard convolution), but this time e.g. if you have a kernel of 2 (in the time domain) from input of size 2 you produce output of size 3. Both left and right frames of output need to be cropped as they are corrupted (by the lack of data). After that operation you get 1 frame of output and to get additional frame for next transposed convolution layer you just concatenating it with output (of that particular layer) of previous inference. For tSTMC we are doing standard convolution in the time axis (and transposed convolution in any other axis). We implemented tSTMC as one (fused) convolution operation where we used convolutional arithmetic that you may find in [1] differently for time dimension then for any other dimension.
>
> > Figures are not described in detail in the main text.
>
> We updated the paper to better describe pictures. Thank you for pointing that out.
>
> > In Table 3, are there any missing numbers or what do the centered numbers mean?
>
> The tool we used did not provide us information about peak memory while using GPU. We decided to not compute this via different method as then this results will not be directly comparable with other results. Also, we mostly foresee uses of our method on smaller computation units like microcontrollers or some more specialized SoCs like NPUs. We might be wrong, but then there is also a question if we do have a lot of constrain regarding peak memory consumption in larger systems.
>
> We updated the paper and write why there are missing values as this was not stated there.
>
> > Quality: Fair. Experiments, especially the causal convolution based ones show significant reduction in inference time but it comes with a 6% degradation of SI-SNRi. Is there a way to avoid this trade-off?
>
> The reason behind SI-SNRi degradation is the absence of 'future' data. As 'future' data we mean the data that is to the right of currently calculated output frame. The possible solution to the problem would be the usage of different number of future frames. We provided an experiment to assess how our model depends on the size of the future data to demonstrate the trade-off (see paragraphs ‘Influence of look-ahead’ in sections 3.1.1 and 4.1. Thank you for suggestion and we hope you will find our results interesting.
>
> The one idea how mitigate this effect without losing causality that comes to our mind is usage of some form of forecasting of future signal to replace the real one. Although this will require some additional computation which will come with a cost in computational complexity.

---

> > ### Author Response · Authors · 2022-11-18
> > **Response 2/3**
> >
> > > Reproducibility: Some experimental details are provided, it should be reproducible at a high level but some details are missing. For example, whether there is model quantization, how the inference time processing is performed exactly: What is meant by ``Input data is processed in chunks of arbitrary size in an online mode.’’?
> >
> > We used default TFlite converter settings for all models without any additional optimization (so we also didn't use quantization). We updated our paper to make it clearer.
> >
> > By 'chunks of arbitrary size' we mean that the size of processed input data within one inference can be changed in time if needed. This process of adaptation of input size is important in real-time processing as the latency of the system can change over time if for example our processor need to do other time consuming tasks. By 'online mode' we mean that we have constant stream of input data.
> >
> > > What is the task? Is it denoising or speech separation? From the dataset (DNS), it sounds like the former but in several paragraphs it is told to be separation.
> >
> > The main task used to illustrate our method in the paper is speech separation (however we think it is conceptually equivalent to speech enhancement/denoising – it depends only on what is considered as the source and target). Additionally, we have also provided results for audio classification. The DNS dataset was used for training because it consists of clean speech samples and separated noise audio that can be conveniently used as target separated audio samples while their mix can serve as the model input.
> >
> > Thank you for suggesting that this might be not clear enough, we put more empathize on it in the corrected version of the paper (actually we even named sections by the task name).
> >
> > > In Appendix A, the equation for b_i does not match with the matrix shown above it. Should the filter be reversed in the matrix $[k_3,k_2,k_1,0,0]$?
> >
> > Please note that the equation uses $a_{i-2}, a_{i-1}, a_i$ and not $a_1, a_2, a_3$. That is why the order is $[k_1, k_2, k_3, 0, 0]$.
> >
> > > Why are the inference time patterns different between Fig 4D and 4E?
> >
> > In 4E we use tSTMC, which does standard convolution in one direction (time) and transposed convolution in other (frequency) as one operation (this operation is shown in Fig 3. and in the paragraph directly under this figure). In Fig. 4D we use STMC in encoder and transposed convolution (in both directions) with simple cropping (to retain causality) of 1 most from the right (most future) frame (without cropping after transposed convolution our output of a layer would be shifted by this one frame - we provided additional context on that in the corrected version of the paper). The highest difference we achieved in first transposed convolution layer, which is due to lack of a stride in frequency axis in that particular layer, which means we do not need to use the data scattering operation.
> >
> > > In Appendix B, there is a mention of quantization but the main paper does not mention anything related to this. Does the paper apply model quantization in the experiments of the main text?
> >
> > We do not use any quantization in any model we present here (or any other optimization method we can disable while compiling TFlite) to avoid any influence of them over our method, although we can confirm that quantization work with our method and does not seem to influence it negatively as we use the quantized version of a model in within the company (not paper related).

---

> > > ### Author Response · Authors · 2022-11-18
> > > **Response 3/3**
> > >
> > > > In the last sentence of the conclusion, it is mentioned that the SI-SNRi degradation in the causal STMC version is attributed to the lack of future context. Could the training vs. inference time mismatch be another reason?
> > >
> > > We have ruled out the training vs. inference time mismatch from the possible causes of SI-SNRi degradation. To prove that, we can compare our models in 2 groups - non causal and causal as in both cases we provided the results for models that uses our method and that does not. We also compared outputs of mentioned models directly and the difference between two whole spectrograms was around 1e-7 (our spectrogram output data for validation set ranges in {-1000, 1000} mean absolute value is 0.1, and average sum over all values of spectrogram is 56958) which is about 2e-10% of relative error. We attributed that difference to numerical stability of convolution as both operations are equal. We also added a note in our newly added ‘Influence of look-ahead’ experiment that this experiment actually prove that.
> > >
> > > As we said before – the corrected version of the paper is ready. For your convenience, as a lot have changed in the paper, here is the list of changes made to the paper:
> > >
> > > - The ASC task was moved from appendix to main text (sections 3.2 and 4.2)
> > > - Added ‘Top-1 Acc’ column to table 3 (sections 3.2 and 4.2)
> > > - Mathematical description of STMC was rework (section 2.1)
> > > - Added ‘STMC with different model sizes’ experiment (sections 3.1 and 4.1)
> > > - Added ‘Influence of look-ahead’ experiment (sections 3.1 and 4.1)
> > > - Added additional content to ‘Efficient causality’ (now in appendix A)
> > > - Figures and tables size was decreased to fit new content
> > > - Parts of introduction were removed to fit new content
> > > - One paragraph from ‘Conclusion’ was removed to fit new content
> > > - Other minor changes
> > >
> > > We believe our work is quite important for AI field and if you find that too please consider changing our recommendation score.
> > >
> > > [1] Dumoulin, V., & Visin, F.. (2016). A guide to convolution arithmetic for deep learning. https://arxiv.org/abs/1603.07285

---

### Official Review · Reviewer_TkYW · 2022-10-25

**Confidence:** 3
**Correctness:** 4
**Technical Novelty And Significance:** 3
**Empirical Novelty And Significance:** 2
**Recommendation:** 5

**Clarity, Quality, Novelty And Reproducibility:**

I have some concern about the clarity of the paper:
- Section 2.1 is quite confusing for me, I don't get how the first and second equation of page 4 are equals, could the authors provide more mathematical details? also please number the equations.
- I think it would be help to also provide the mathematical description of standard convolution layer.

About the experiments:
- The experiments are quite limited, I want to see the results of MoViNets and of similar related works to be able to judge the benefit of the proposed approach.
- Simple baselines with very low latency are also missing. I think it's not hard to design a LSTM or CNN with 10ms latency (just one frame), it would be helpful to see the trade-off in performance there.
- I'm a bit confused by how the inference time is computed, is it computed over the whole testset? Please clarify. It would be useful to also report the Real Time Factor or equivalent (the time it takes to compute an input of a given duration) as this is more important in real-time application.
- Why is the processing time included in the latency? In my experience it doesn't affect the latency of real-time systems as long as the processing time is shorter than real-time (i.e. shorter than 10ms to process 10ms of input) for instance.
- Do the authors have an intuition about why the mobile is 5x faster than the desktop? It looks counter-intuitive as the desktop has more processing power.
- The GhostNet experiments in the appendix are quite interesting, would it be possible to add them to the main text? I think they are strengthening the paper.

The proposed approach seems novel, and the findings are significant for real-time audio applications.

In terms of reproducibility, the proposed method would be difficult to reproduce because of the vague mathematical description and the absence of shared code.


**Strength And Weaknesses:**

Strengths:
- The paper is well structured and easy to read
- The proposed approach seems novel AFAICT and well situated w.r.t. previous works.
- The experiments shows the benefit of the proposed layer for real-time applications

Weaknesses:
- The description of the proposed layer is not very clear (see next question)
- The experiments are a bit limited.
- The task is not necessarily the most representative for audio and speech application, more experiments on other tasks such as ASR or VAD would have been welcome.

**Summary Of The Paper:**

This paper introduces the Short-Term Memory Convolution (STMC) layers, which is designed to be a faster version of the convolution layers (in terms of computation time) and also achieve lower latency for real-time application. The paper first presents the approach, based on shift registers. The proposed layer is then evaluated on an audio separation task and compared with baseline models. It is shown to yield similar performance than the baseline with significant gain in inference time and latency.

**Summary Of The Review:**

This paper Introduce a novel layer which looks useful for audio real-time applications. But in the current form, the paper is not clear enough on the methods and the experiments are not bad but seems limited. Hence I'll recommend rejection, but I would be happy to revise if the authors can adress my concerns.

---

> ### Author Response · Authors · 2022-11-18
> **Response 1/3**
>
> Hello! As all of reviews we received were extremely useful and helped us improve our paper, we would like to thank you for your time and insight. We are happy to say that the corrected version of the paper is ready but before you have a look please let us answer all of your questions and concerns.
>
> > The task is not necessarily the most representative for audio and speech application, more experiments on other tasks such as ASR or VAD would have been welcome.
>
> It is important to note that our method is task or model agnostic (as long as it follows described limitations in section 2.2 Limitations). In paper we also showed the results for audio classification (which is similar to ASR or VAD), where we used a GhostNet (this results will be moved to the main text following your suggestion). The task of speech separation is our current field of interest and that's why it is used as a main task in the paper. The results achieved with STMC might differ as the architecture and size of the model will have an impact, but it will improve every CNN as long as it does not contain only one layer. We wanted to present it as a generalized method that can be used in every CNN and not as a concrete architecture for specific task (like in MoViNets), as we know that it can be used outside our domain and we are looking forward to see what other researchers do with it.
>
> > Section 2.1 is quite confusing for me, I don't get how the first and second equation of page 4 are equals, could the authors provide more mathematical details? also please number the equations.
>
> We updated section 2.1 as it seemed to be confusing and we have numbered equations.
>
> > I think it would be help to also provide the mathematical description of standard convolution layer.
>
> Thank you for suggestion! We provided the mathematical description of standard convolution in the corrected version of the paper. Please see eq. (1) and its description.
>
> > The experiments are quite limited, I want to see the results of MoViNets and of similar related works to be able to judge the benefit of the proposed approach.
>
> We see STMC as a generalization of the stream buffering technique from MoViNets. The stream buffers from MoViNets were applied specifically to standard convolutions. STMC expands this technique to strided convolutions, transposed convolutions, and skip connections which are present in U-nets, but also these type of operations can be used to build any convolutional model. Therefore, from a theoretical point of view the results on simple convolutional models should be the same for STMC and stream buffers. If there are any differences, they would be due to implementation details. Our aim was to focus on the generalizability of the method and not on a specific implementation. It is also worth mentioning, that the implementation of stream buffers from MoViNets is probably slightly different than STMC for simple convolutions, because using the stream buffers slightly reduces the accuracy of the model, while STMC does not affect the results. In addition, adding stream buffers to the model slightly increases the computational complexity (GFLOPS), which we do not understand. STMC, thanks to buffering, reduces the computational complexity by a factor of 5 in our case. The implementation details of stream buffers are, however, not well explained in the paper (Kondratyuk et al. [21]).
>
> We hope that our experiments on inference time reduction and the presented results will motivate the community to start the discussion on the subject and to further apply and test STMC to many other model architectures.
>
> > Simple baselines with very low latency are also missing. I think it's not hard to design a LSTM or CNN with 10ms latency (just one frame), it would be helpful to see the trade-off in performance there.
>
> We hope that our audio classification experiments will serve that purpose. Initially, we didn't provide any metrics making the improvements hard to judge. We updated Table 3 to reflect that. We also added the experiment where we measure the improvement of inference time in the function of model size for our U-Net model (see paragraphs ‘STMC with different model sizes’ in sections 3.1.1 and 4.1, which hopefully will give some intuition on what improvements can be expected from different model sizes. Thank you for a suggestion and we hope you will find this addition interesting.

---

> > ### Author Response · Authors · 2022-11-18
> > **Response 2/3**
> >
> > > I'm a bit confused by how the inference time is computed, is it computed over the whole testset? Please clarify. It would be useful to also report the Real Time Factor or equivalent (the time it takes to compute an input of a given duration) as this is more important in real-time application.
> >
> > Inference time was calculated by measuring time needed to calculate 1 000 inferences (after warmup of 32 inferences) of randomized input signal (1 000 different input samples). We reported the average time of this 1 000 inferences. Added simple note that our models have RTF under or equal to 1. Thank you for suggestion. In the paper we don’t see much point in showing what RTF we achieved on all models as it depends on input length (which might be changed) and actually with RTF you just hit desired value (which is model, application and system dependent). For your information, here are our RTF values for each model (by assumed input signal length for latency estimation):
> >
> > | Platform | Model      | RTF |
> > | ----------- | ----------- | ----------- |
> > | | U-Net      | 0.468 |
> > | | INC | 1.000 |
> > | | STMC | 0.997 |
> > | Desktop  | C. U-Net | 0.328 |
> > | | C. INC | 1.000 |
> > | | C. STMC | 0.627 |
> > | | C. tSTMC | 0.609        |
> >
> > | Platform | Model      | RTF |
> > | ----------- | ----------- | ----------- |
> > | | U-Net      | 0.095 |
> > | | INC | 1.000 |
> > | | STMC | 0.804 |
> > | Mobile | C. U-Net | 0.088 |
> > | | C. INC | 1.000 |
> > | | C. STMC | 0.803 |
> > | | C. tSTMC | 0.779        |
> >
> > We believe that if RTF is satisfied the value of latency is far more important as RTF does not change much at this point. We hope you will find our reasoning satisfactory.
> >
> > > Why is the processing time included in the latency? In my experience it doesn't affect the latency of real-time systems as long as the processing time is shorter than real-time (i.e. shorter than 10ms to process 10ms of input) for instance.
> >
> > To compute the output we need to acquire the whole signal needed for overlap-add (we discarded influence of overlap-add in our calculations and simplify it to just a size of input data, as this is not a part we are optimizing). The last computation starts after the last frame is collected and to send it to the output buffer we need to do this computation. Computation of other frames needed for overlap-add is discarded as it is done while next frame is collected. Whole computation for one frame includes inference time and time needed to calculate STFT/iSTFT. We updated our latency description in the corrected version of the paper to make it more clear (see paragraph ‘Evaluation on desktop’ in section 3.1.1)
> >
> > > Do the authors have an intuition about why the mobile is 5x faster than the desktop? It looks counter-intuitive as the desktop has more processing power.
> >
> > We also thought that when we saw the results as this indeed seems counter-intuitive, but after some discussion we came to the conclusion that TFlite is designed for embedded devices and that probably it is not as efficient on x86 platform. We also think in such small target inference time there is a large influence of memory access, which is faster on ARM because the model is stored directly on silicon.
> >
> > > The GhostNet experiments in the appendix are quite interesting, would it be possible to add them to the main text? I think they are strengthening the paper.
> >
> > Thank you for pointing that out. We think that too. Initially we put them into appendix as we run out of space and did not want to make it too crowded. We changed the structure of paper and we added this results to main text. We also added the column with an accuracy which should make it easier to compare models.
> >
> > However, please be aware that this change required us to remove some other parts of paper (mostly from Introduction) to fit 9-page hard limit of ICLR. If limit increases we will happily bring them back as we think those part were valuable as well.
> >
> > > The proposed approach seems novel, and the findings are significant for real-time audio applications.
> >
> > We also would like to add that this research enables us to go even further in computation complexity reduction research and to produce other novel approaches, which will hopefully be published shortly after this one. We hope that this work will inspire other researchers to create similar solutions aimed at increasing the efficiency of DNN models. Currently, there is a big trend on increasing the model size on the pursuit for better metrics and often the efficiency is overlooked. To put that into perspective Xu et al. stated that 'The human brain is more than five orders of magnitude more energy efficient than all current DNNs' [1]. Whereas in may task we already can achieve superhuman performance, we are far away from human-level efficiency.

---

> > > ### Author Response · Authors · 2022-11-18
> > > **Response 3/3**
> > >
> > > > In terms of reproducibility, the proposed method would be difficult to reproduce because of the vague mathematical description and the absence of shared code.
> > >
> > > As it was stated before we updated our mathematical description. Sadly, we cannot provide the code for our method as it has proprietary parts, but we would like to make our method as easy to implement as we can so if you have additional suggestions how to upgrade it, please let us know! If our inner politics regarding code change, we will happily provide the code.
> > >
> > > As we said before – the corrected version of the paper is ready. For your convenience, as a lot have changed in the paper, here is the list of changes made to the paper:
> > >
> > > - The ASC task was moved from appendix to main text (sections 3.2 and 4.2)
> > > - Added ‘Top-1 Acc’ column to table 3 (sections 3.2 and 4.2)
> > > - Mathematical description of STMC was rework (section 2.1)
> > > - Added ‘STMC with different model sizes’ experiment (sections 3.1 and 4.1)
> > > - Added ‘Influence of look-ahead’ experiment (sections 3.1 and 4.1)
> > > - Added additional content to ‘Efficient causality’ (now in appendix A)
> > > - Figures and tables size was decreased to fit new content
> > > - Parts of introduction were removed to fit new content
> > > - One paragraph from ‘Conclusion’ was removed to fit new content
> > > - Other minor changes
> > >
> > > We believe our work is quite important for AI field and if you find that too please consider changing our recommendation score.
> > >
> > > [1] Xu, X., Ding, Y., Hu, S.X. et al. Scaling for edge inference of deep neural networks. Nat Electron 1, 216–222 (2018). https://doi.org/10.1038/s41928-018-0059-3

---

### Official Review · Reviewer_xgZF · 2022-10-31

**Confidence:** 3
**Clarity, Quality, Novelty And Reproducibility:** Clerity and quality are good. All res…
**Correctness:** 4
**Technical Novelty And Significance:** 2
**Empirical Novelty And Significance:** 3
**Recommendation:** 6

**Strength And Weaknesses:**

Overall the paper is a reasonable contribution. Although the direct technical novelty is minimal there is quite an empirical value to the observations made in the STMC/tSTMC setups. The paper is well written and easy to understand.

Questions/Weaknesses?
1) The authors have already mentioned that the proposed models' efficacy is dependent on the depth of the network and losing gains as depth increases. On this note can the authors confirm two things: (a) For networks with strided convolutions we cannot expect to see such gains since the buffer size increases drastically (any idea/estimate how bad this can be?) and (b) the latency calculations included times for stft/istft along with other, but given the simplest case of memory-1 buffer shown here in this paper these two times are fixed across all the models (rows in table 1)? Or is it not, am I missing something basic here!
2) Does the TFlite model do any optimizations that may have inadvertant influence on the estimates shown in Table 2? i.e., trying to better disentangle the influence of the proposal vs. implicit optimization of the packages?
3) What was the rationale for using U-Net based models for benchmarking as opposed to classical CNNs for other types of CV/AV tasks?

**Summary Of The Paper:**

The authors identify an implicit characteristic of convolutional layer operations and exploit that to show benefits in operational speed/efficiency for CNN transposed CNN layers and pooling layers in deep nets. The idea itself is neat and clear. And the paper is presented well. The authors evaluate the proposed on non-causal and causal formulations of U-Net.

**Summary Of The Review:**

I believe this is a reasonable contribution. The authors acknowledge the limitations of the work which is appreciated.
Nevertheless, the evaluations are minima and some of the questions posed above are to be addressed to get clarity on the choices made in the proposal.

---

> ### Author Response · Authors · 2022-11-18
> **Response 1/2**
>
> Hello! As all of reviews we received were extremely useful and helped us improve our paper, we would like to thank you for your time and insight. We are happy to say that the corrected version of the paper is ready but before you have a look please let us answer all of your questions and concerns.
>
> > The authors have already mentioned that the proposed models' efficacy is dependent on the depth of the network and losing gains as depth increases. On this note can the authors confirm two things: (a) For networks with strided convolutions we cannot expect to see such gains since the buffer size increases drastically (any idea/estimate how bad this can be?) and (b) the latency calculations included times for stft/istft along with other, but given the simplest case of memory-1 buffer shown here in this paper these two times are fixed across all the models (rows in table 1)? Or is it not, am I missing something basic here!
>
> It is actually another way around! In section 2.1 (3.1 before) we wrote: “It is worth noting that the effectiveness of STMC increases with the depth of the network”. Although, if you have a look at our results from newly added paragraph ‘STMC with different model sizes’ in 4.1, you will find out that if we increase the number of kernels within the network, the relative improvement in peak memory consumption decreases with model size (and relative inference time improvement stays the same).
>
> (a) The inference time for strided convolutions is also optimized in similar manner as in standard convolutions when we use STMC. The difference is in the number of stored states of the network (number of shift registers) at particular time point, but not all states will be used at single inference. Even though the improvement in inference time for strided convolution is the same as in the standard one, it increases the peak memory consumption of the network as the number of additional states follows $2^x$ formula, where x is the number of strided convolutions before the layer where we calculate states. E.g. if we have 4 convolutional layers and 2 first are strided the number of states for those layers will be {1, 2, 4, 4}. We will also need to remember about the buffer for current frames. For kernels of size 2 in the time domain the buffer for current states is the same size as for single past state and for larger kernels the influence of current buffer in relation to past buffers diminishes. We also targeted this problem in our recent research (which we actually concluded several days ago), but it turns out to be a quite large and became a separate method which uses STMC to work, so we are planning to release it as a follow up paper after this one. We also have 2 new 'branches' for this research (both with working PoC!) so this work may be seen as a first of a series. We hope other researchers will be inspired by this work as much as we are.
>
> (b) Tables 1-3 consist latency (which includes STFT/iSTFT among other components as described in section 3.1.1) and inference time, which is one of the components affecting the latency. We updated our paper to elaborate more on how we calculate latency as it seemed to be confusing. We added 1 new paragraph at the end of 1 Introduction (just before 1.1 Related works) and also changed the ‘Evaluation on desktop’ paragraph in 3.1.1 where our latency calculations is described.
>
> > Does the TFlite model do any optimizations that may have inadvertant influence on the estimates shown in Table 2? i.e., trying to better disentangle the influence of the proposal vs. implicit optimization of the packages?
>
> We used default TFlite conversion in all our models without any additional optimizations (like quantization). Because, all models were converted in the same manner we believe they should be directly comparable with each other. As far as we know TFlite for default settings (w/o optimization) removes only the parts of model not needed for inference (like backprop related things).
>
> We saw also similar improvements when we did not compile our models to TFlite but for online systems we mostly use converted models (as they are much faster than TF models) and that is why we report this models instead of TF ones.

---

> > ### Author Response · Authors · 2022-11-18
> > **Response 2/2**
> >
> > > What was the rationale for using U-Net based models for benchmarking as opposed to classical CNNs for other types of CV/AV tasks?
> >
> > Mainly, in our research we targeted real life problem that exist for small appliances - speech enhancement (e.g used for phone talks or headphones with acoustic noise cancellation). In this task U-Net is a standard and well known architecture but it is often too computational heavy for those devices. Also, U-Net utilizes both transposed convolutions and skip connections which, as far as we know, were not considered from the latency minimization point of view by other authors. Moreover, audio processing is our main field of research, so it was much easier for us, as we have a good intuition in that domain. We also did experiments with audio classification models (the results are included section 4.2).
> >
> > We believe that our method is task and model agnostic but is limited by several constrains described in section 2.2 Limitations. That being said, we think that the task and model we used to show STMC is not the most important.
> >
> > As we said before – the corrected version of the paper is ready. For your convenience, as a lot have changed in the paper, here is the list of changes made to the paper:
> >
> > - The ASC task was moved from appendix to main text (sections 3.2 and 4.2)
> > - Added ‘Top-1 Acc’ column to table 3 (sections 3.2 and 4.2)
> > - Mathematical description of STMC was rework (section 2.1)
> > - Added ‘STMC with different model sizes’ experiment (sections 3.1 and 4.1)
> > - Added ‘Influence of look-ahead’ experiment (sections 3.1 and 4.1)
> > - Added additional content to ‘Efficient causality’ (now in appendix A)
> > - Figures and tables size was decreased to fit new content
> > - Parts of introduction were removed to fit new content
> > - One paragraph from ‘Conclusion’ was removed to fit new content
> > - Other minor changes
> >
> > We believe our work is quite important for AI field and if you find that too please consider changing our recommendation score.

---

### Decision · Program_Chairs · 2023-01-20

**Decision:**

Accept: poster

**Justification For Why Not Higher Score:**

- Evaluation can be considered non-conventional.
- Code is not provided. If provided it can add more value to the submission and its long-term impact.

**Justification For Why Not Lower Score:**

Solid work with clear practical benefits. The presented work deserves to be accepted as a poster at the venue.

**Metareview: Summary, Strengths And Weaknesses:**

I Summary:

- I.1 Investigated Problem:
The paper investigated the problem of latency minimization, memory consumption, and inference time reduction for real-time processing of time series signals as it is a critical issue for many real-world applications

- I.2 Proposed Solution:
The authors present Short-Term Memory Convolution (STMC) and its transposed (tSTMC) counterpart layers. The proposed solution relies on the buffering of the layer's outputs. The proposed layers introduce an additional cache memory to store past observed outcomes. The cache memory provides values calculated in the past to subsequent calls of the STMC layers in order to calculate the final output without any redundant computations. When processing a time series all calculations are performed exactly once

- I.3 Validity Proof of the Proposed Solution:
Through an application of the solution to a U-Net model for an audio separation task. Several variants of a U-Net architecture were compared, differing in causality and buffering types.
 A 5-fold reduction of inference time was achieved compared to incremental inference, which is considered the state-of-the-art method for online processing using convolutional models. All the causal models used in the test achieved approximately a 6% reduction in SI-SNRi compared to their non-causal counterparts. The cause of the score reduction is attributed to the lack of future context and not the used caching method.

II Strengths:

- II.1 From a structural point of view:
 . The paper is well-structured and easy to understand.
- II.2 From an analytical point of view:
. Fair experiments have been conducted to demonstrate the potential of the proposed method;
  .The Intellectual honesty demonstrated by the authors is appreciated by the reviewers as some limitations of the proposed model have been clearly mentioned;
 . Significant empirical evidence is provided to support the case of the proposed solution in terms of reduction of inference time and memory consumption;
- II.3 From a perspective of soundness (development, unity, and coherence) and completeness (correctness):
 . Active interaction of the authors during the rebuttal period and their openness to concerns and questions raised by the reviewers allowed them to clarify certain points.

III Addressing what can be thought of as weaknesses

Before the rebuttal, some of the reviewers raised concerns about:
	1- the clarity of the proposed method:
		The authors updated the submission by editing the section related to the latency description.
	2- the nature of the task related to the conducted experiments:
		Whether it is about source separation or denoting. Authors explained during the rebuttal that the task of speech separation is equivalent to speech enhancement/denoising as secondary sources of signals can be considered as noise to the original source

IV. Potential of the paper:

- IV.1 From a Potential perspective (Potential of the paper to the community): The proposed solution has the potential to be of benefit to the whole community, especially the one interested in real-time processing of time series. The extension and investigation of other models can also be of great benefit to the machine learning community in general.


**Note From Pc:**

if the above contains the word "oral" or "spotlight" please see: "oral" presentation means -> notable-top-5% and "spotlight" means -> notable-top-25%. As stated in our emails, we are disassociating presentation type from AC recommendations

**Summary Of Ac-Reviewer Meeting:**

- During the meeting, reviewers expressed their appreciation for the quality of the work presented as the authors conducted solid experiments and addressed raised concerns and questions asked by the reviewers. The clarity of the presented method and its effectiveness are also appreciated by the reviewers.

- Evaluation of the method was also discussed during the meeting. It is worth mentioning that conducted evaluation is correct and demonstrates the effectiveness of the proposed solution, yet it can be considered non-conventional.
Clarification of the equivalence between source separation and the denoising task has been discussed during the meeting. Reviewers provided an explanation for their assessment of equivalence between both tasks. It is worth mentioning that such an assessment may induce readers in error, especially the ones familiar with audio-related tasks. Nevertheless, the explanation of the authors during the rebuttal is appreciated.

- The community can benefit from the proposed method as it has practical advantages. We understand that has proprietary parts, yet it would be of great benefit to the machine-learning community if source code and implementation details are provided. Pseudo-source code details are also appreciated.